# Assembly dynamics of FtsZ and DamX during infection-related filamentation and division in uropathogenic *E. coli*

Bill Söderström [1✉], Matthew J. Pittorino [1], Daniel O. Daley[2] & Iain G. Duggin [1]

During infection of bladder epithelial cells, uropathogenic *Escherichia coli* (UPEC) can stop dividing and grow into highly filamentous forms. Here, we find that some filaments of *E. coli* UTI89 released from infected cells grow very rapidly and by more than 100 μm before initiating division, whereas others do not survive, suggesting that infection-related filamentation (IRF) is a stress response that promotes bacterial dispersal. IRF is accompanied by unstable, dynamic repositioning of FtsZ division rings. In contrast, DamX, which is associated with normal cell division and is also essential for IRF, is distributed uniformly around the cell envelope during filamentation. When filaments initiate division to regenerate rod cells, DamX condenses into stable rings prior to division. The DamX rings maintain consistent thickness during constriction and remain at the septum until after membrane fusion. Deletion of *damX* affects vegetative cell division in UTI89 (but not in the model *E. coli* K-12), and, during infection, blocks filamentation and reduces bacterial cell integrity. IRF therefore involves DamX distribution throughout the membrane and prevention of FtsZ ring stabilization, leading to cell division arrest. DamX then reassembles into stable division rings for filament division, promoting dispersal and survival during infection.

[1] Australian Institute for Microbiology and Infection, University of Technology Sydney, Sydney 2007 NSW, Australia. [2] Department of Biochemistry and Biophysics, Stockholm University, Stockholm 106 91, Sweden. ✉email: bill.soderstrom@uts.edu.au

It is estimated that some 150 million people are affected by urinary tract infections (UTIs) each year[1]. The severity of these infections ranges from minor to life-threatening[2–4]. While there are many different species of bacteria that can cause UTIs, the major causative infectious agent is uropathogenic *Escherichia coli* (UPEC), which is responsible for more than 80% of reported cases[5].

The UPEC infection cycle is initiated with bacteria being endocytosed by epithelial cells in the human urinary tract[6]. Once inside the host cells, the bacteria proliferate and undergo morphological changes to adopt cocci-like shapes while forming biofilm-like intracellular bacterial communities (IBC)[7]. These IBCs are exposed to various intercellular cues and develop into large communities that dominate the host cell. Eventually, the colony ruptures the host cell and disperses into the surrounding environment. During dispersal, a subset of the bacteria may stop dividing and grow into long filaments, some hundreds of microns long. The filaments can later divide back into rods and reinitiate the infection cycle[8]. Here we use the term infection-related filamentation (IRF) for this distinctive type of bacterial response.

It is not clear why UPEC undergoes IRF, but similar morphological changes are part of bacterial stress responses[9–11]. It has been suggested that filamentation might be a survival strategy to evade the immune system[12]. It is clear, however, that in order to reinitiate the infection cycle and colonize new host cells, the filaments must revert back to their typical rod shape[13]. As such, the reversion of filaments to rods is a crucial step in the UPEC infection cycle. Very little is known about how UPEC filaments reinitiate division to revert to rods.

During vegetative growth, binary fission is mediated by the bacterial cell division machinery—the divisome[14]. This multi-protein nanomachine is organized by the highly conserved FtsZ protein[15]. Together with a number of other early arriving proteins (e.g., the essential FtsA and ZipA[16,17]) FtsZ forms a proto-ring at the midcell when cells are primed to divide[18]. In a second step, the divisome matures by recruiting another set of core divisome proteins that have vital functions in chromosome segregation and peptidoglycan (cell wall) synthesis[19]. Apart from the core proteins, *E. coli* has specialized divisome-associated proteins that modify the septal peptidoglycan during division via a SPOR (*Sporulation-related repeat*) domain[20,21].

SPOR domains are widely conserved and have been found in more than 1500 bacterial genomes[22]. *E. coli*, including UPEC, has four SPOR-containing proteins[22], DamX, DedD, FtsN and RlpA, which localize to the division septum by binding to denuded glycan strands via their SPOR domains[20,22]. The molecular function(s) of the SPOR proteins in pathogenic bacteria are yet to be fully understood but important observations have been made during growth in standard laboratory conditions. FtsN, the only essential SPOR-domain protein, is thought to trigger division constriction[23,24]. RlpA is a lytic transglycosylase important for shape and division in *Pseudomonas aeruginosa* and *Vibrio cholerae*, but in *E. coli* the importance of this function is not clear[25,26]. DamX and DedD stimulate the enzymatic activity of cell wall synthases in *E. coli*[27]. Importantly, *damX* gene expression was found to be upregulated during the dispersal (filamentation) stage of bladder cell infection, and DamX is essential for UPEC IRF[28].

In the present study, we have established an approach for studying UPEC filamentation and filament division using high-resolution and time-lapse fluorescence microscopy. We have used this approach to quantitatively characterize the growth and division dynamics of UPEC filaments dispersed from cultured bladder cells. To gain a better understanding of the dynamics and behaviour of the cell division machinery in IRF, we also visualized two divisome proteins expected to be important, FtsZ and DamX.

Interestingly, FtsZ dynamically assembled in transient ring-like structures in filaments, which differs from normal rod cell division where one ring assembles and then completes division. On the other hand, DamX assembly into a ring always resulted in constriction of the membranes followed by division of the UPEC filaments. These data enable us to discriminate between two alternative models for the function and localization of DamX in UPEC filaments (post-infection). These models propose that DamX: (1) is localized at potential division sites along the filament during the arrest of cell division for filamentation, and then switches its function at those sites to promote division during reversal, or (2) is delocalized as part of the filamentation mechanism, and then assembles as rings, like in rod cell division, for the filament division process. The results consistently supported the second model.

## Results

**Shorter filaments are more likely to revert to rods**. We used an in vitro urinary tract infection model system[29], based on infection of cultured human bladder epithelial cells, to generate filaments of the model UPEC strain UTI89[30]. We transformed UTI89 with plasmid pGI5[29], which constitutively produced cytoplasmic sfGFP and did not alter growth rate, cell size or shape in standard laboratory conditions, i.e., during growth in rich medium at 37 °C (Supplementary Fig. 1). This strain was used interchangeably with wild-type (WT) UTI89 where appropriate. After infection of bladder epithelial cells with UTI89/pGI5, and then 20–22 h exposure to a flow of urine to induce filamentation and dispersal, a sample of the flow-through containing material from ruptured bladder cells was collected, washed once in PBS, resuspended in LB, immobilized on agarose pads, and imaged with light microscopy. As observed in previous studies, there was a mixture of both filaments and rods (Fig. 1a)[13,29] and some areas of the slide showed noticeable clusters of filaments (Fig. 1b). We defined a cell as being a filament if it was at least two and a half times longer than the average length of cells grown in liquid LB culture. Using this definition, a filament was ~8 μm or more in length. The substantial heterogeneity in filament lengths (Fig. 1b, c) was consistent with the expected asynchronous dispersal of UPEC from the bladder cells[29].

To characterize the growth and division dynamics of UPEC filaments, dispersal samples were placed on a LB-agarose pad and then time-lapse microscopy movies were captured with one image acquired every 3–5 min for at least 120 min. We observed several outcomes in the filaments. Relatively shorter filaments often grew into longer filaments (>20 μm) before starting to revert to rods, whereas some filaments elongated and then suddenly lysed (Supplementary Movie SM1). In contrast, many of the initially longer filaments (>50 μm) did not elongate further (Supplementary Movie SM2). Most of the shorter filaments (<50 μm) reverted to multiple rods over the course of imaging (Supplementary Movie SM3).

To quantify the state of filaments, we classified the fates of 343 filaments (from three different infection experiments) over at least 120 min into one of three groups: (1) 'viable', if they divided at least once, (2) 'Lysed during imaging', indicating filaments that did not divide and lysed during the image acquisition time (Supplementary Movie SM4), and (3) 'dead', representing shells of filaments that were highly translucent and did not divide nor extend in length at all during the imaging period. Supplementary Fig. 2 shows examples of 'viable' and 'dead' filaments. Average lengths of filaments at the start of imaging and their classification frequencies were: viable = $56.3 \pm 56.5$ μm ($n = 208$, ~61%), growing but no division = $117.6 \pm 73.4$ μm ($n = 67$, ~20%) and dead = $122.6 \pm 79$ μm ($n = 68$, ~20%) (Fig. 1c). Therefore, most

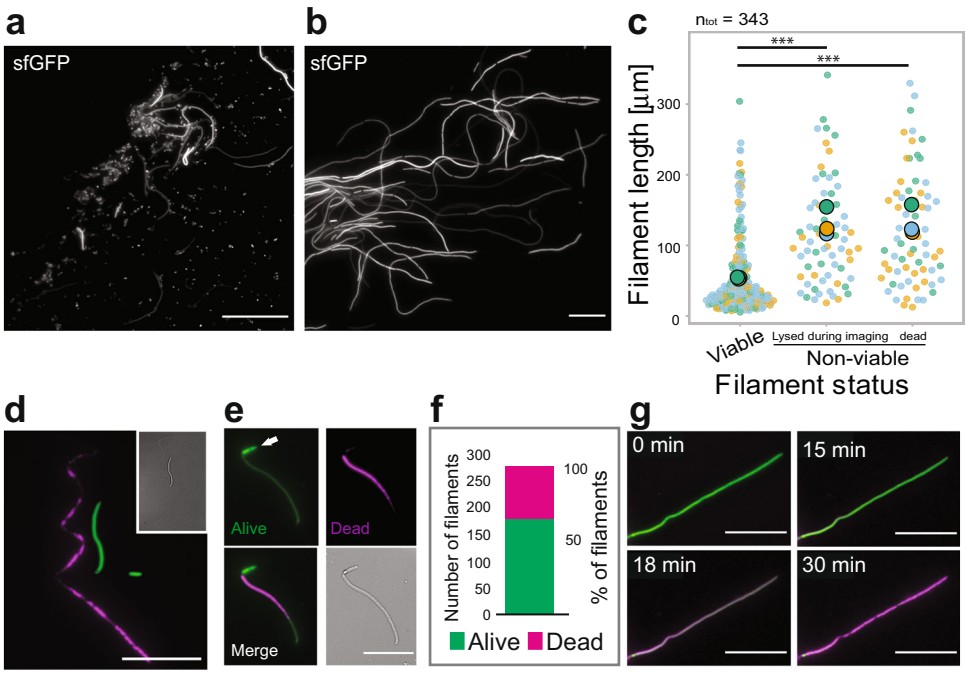

**Fig. 1 Shorter filaments are more likely to be viable and revert to rod cells. a** Fluorescence microscopy image showing a mixture of sfGFP-expressing short cells and filaments released from bladder cells after urine exposure. **b** Length heterogeneity in a cluster of filaments. **c** Shorter filaments are more likely to be viable after a round of infection. A total of 343 filaments were analysed from time-lapse imaging of three different infection experiments (Blue, Green and Yellow dots, respectively). Large, coloured dots represent the average of the respective experiment. Only cells classified as "filaments" (i.e., equal to or longer than 8 μm) were included in this analysis. Overall average filament lengths were 56.3 ± 56.5 ($n = 208$) for viable, 122.6 ± 79 ($n = 68$) for dead, and 117.6 ± 73,4 ($n = 67$) for dying (mean ± SD). *P* values are from unpaired two-tailed *t* tests. ***$P < 0.0001$. 95% confidence interval. **d** WT UTI89 filaments were differentially stained to assess viability (LIVE/DEAD, green and magenta, respectively). **e** Filaments that divided at least once during the first 120 min were classified as viable (white arrow). **f** A total of 275 filaments from three infection experiments were analysed with LIVE/DEAD staining. **g** A representative dying filament, transitioning from green to magenta over time (see Supplementary Movie SM5). Scale bars **a** = 40 μm, Rest = 20 μm. Source data are provided as a Source data file.

of the viable filaments were significantly shorter than the average, indicating that extensive elongation carries a significant risk to filament survival.

To further investigate filament viability, we used a LIVE/DEAD differential staining assay (Supplementary Fig. 3) in which live cells are distinguished by their uptake of the SYTO9 green dye whereas only dead cells take up the Propidium Iodide (PI) red dye (Fig. 1d, e, PI shown in magenta). We examined 275 UTI89 filaments from 3 independent infections and found that 36% stained dead (Fig. 1f), similar to the proportion of non-dividing and dead filaments classified above (~39%). We were also able to follow filaments dying over time, transitioning from green to magenta (Fig. 1g and Supplementary Movie SM5). From here on, we only characterized filaments that were classified as viable.

**Filaments divide with increasing frequency during reversion to rods.** When further analysing the time-lapse movies of the UTI89 filaments, we noticed large differences in the time it took for filaments to initiate division and revert to rods, i.e., the time from placing filaments on the agarose pads until the first division event. This varied from just a few minutes to more than 2 h in the same sample, suggesting a large heterogeneity in metabolic states between filaments. Moreover, filaments elongated by <1 μm, to >140 μm before the first division (mean (±SD) = 19.78 ± 22.3 μm, $n = 101$, Fig. 2a, b). The time-averaged elongation rate ($\Delta L/\Delta t$) varied from 0.08 μm min$^{-1}$ to 1.76 μm min$^{-1}$ in some filaments, with an overall mean of 0.55 ± 0.4 μm min$^{-1}$ (Fig. 2c). For comparison, UTI89 showed an elongation rate of 0.1 μm min$^{-1}$ in standard LB culture (Supplementary Fig. 5), which is similar to the well-studied K-12 strain. We then estimated the specific

growth rate (i.e., elongation rate normalized by length, Fig. 2d) to see if the elongation rate correlates with the starting length of the filaments, as would be expected for homogeneous growth of the filament population. However, we observed poor correlation (Fig. 2e), consistent with a large heterogeneity of metabolic states between filaments (Fig. 2a, d). There was no apparent correlation between the length of a filament at the start of imaging and the time to first division (Fig. 2f).

We next monitored individual filament division over four consecutive generations, after which most cells became too crowded and moved out of focus (Supplementary Movie SM6). Most filaments grew initially (Fig. 2b) and then often divided towards one end, resulting in rod cells 'pinching-off' (Fig. 2g and Supplementary Movie SM6). The sequential generations of the 'mother' filament occurred at a progressively greater frequency (Fig. 2h), often appearing at multiple locations near both ends. This was also be seen by measuring the mean filament interdivision time (Fig. 2j), where, for example, there was a ~25% difference between filament division frequencies after the second and the fourth cells had pinched off (second cell, $\bar{X}_{2nd} = 20.83$ min, compared to the fourth cell, $\bar{X}_{4th} = 15.47$ min). The short interdivision times would suggest that multiple potential division sites may be primed as soon as—or even before—the previous generation has pinched off from the mother filament. This is similar to division frequency during recovery from antibiotic-induced filamentation[31].

The size of newborn cells pinching off from the filament was quite consistent over time, with mean lengths of 4.66 ± 2.32 μm for the first cell, and then 4.89 ± 1.92 μm, 5.05 ± 2.59 μm, and 5.08 ± 3.1 μm for subsequent generations. There was a noticeable

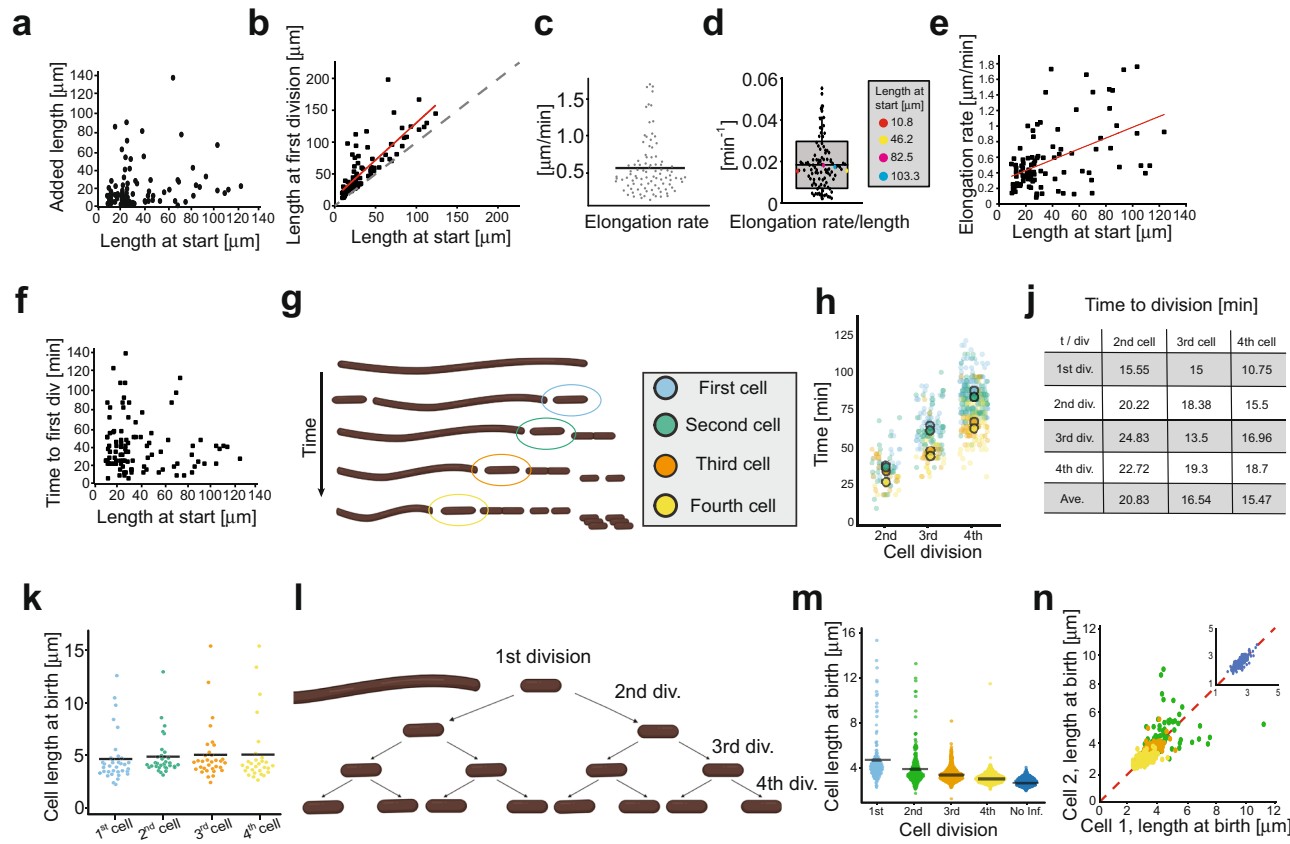

**Fig. 2 Growth and division dynamics of UTI89 filaments that revert to rods. a** Added cell length from the start of imaging to the first division (mean = 19.78 ± 22.3 μm, $n = 101$) vs cell length at the start. **b** Relationship between filament length at start and at first division. The red line indicates a linear fit to the data. Dotted line indicates no growth before first division. **c** Variation in elongation rate between filaments (between 0.08 and 1.76 μm min$^{-1}$). The mean elongation rate was $\Delta L/\Delta t = 0.55 \pm 0.4$ μm min$^{-1}$, $n = 101$. **d** Elongation rate (0.0187 ± 0.0113 min$^{-1}$, $n = 101$) of filaments normalized to their length at the start of the imaging. Box outlines indicate SD, midline indicates mean, whiskers indicate 95% interval. Data maxima and minima are 1.76 and 0.08 min$^{-1}$, respectively. Box to the right high-lights four randomly picked filaments with rates close to mean (red = 0.015 min$^{-1}$, yellow = 0.0152 min$^{-1}$, magenta = 0.0178 min$^{-1}$, blue = 0.017 min$^{-1}$), these filaments ranged from 10 to 100 μm in length, indicating weak correlation between length and elongation rate. **e** The elongation rate was not correlated to the length of the filaments at the start of imaging. The red line shows a linear fit to the data. $R^2 = 0.245$. **f** No apparent correlation between filament length at the start of imaging and time to first division was found. **g** Schematic representation of the first to fourth generations of cells during filament division. **h** Timing of subsequent division events in 'mother' filaments after birth of the newborn cells depicted in **f** (colour key). Larger circles represent the means ($n_{div} = 471$). **j** Mean inter-division times in subsequent divisions of the mother filament after the second, third, and fourth cells have pinched off. Overall means were: $\bar{X}_{2nd} = 20.83 \pm 3.99$ min., $\bar{X}_{3rd} = 16.54 \pm 2.75$ min., $\bar{X}_{4th} = 15.47 \pm 3.41$ min. (average ± S.D.) **k** Mean cell lengths of the smaller newborn cell pinched off from a filament ($n = 135$). **l** Schematic representation of the subsequent division events of pinched-off newborns. **m** Lengths at birth of daughter cells from first to fourth generations. **n** Symmetry of division in newborns. Second generation (green) cells divide more asymmetrically than third (orange) and fourth (yellow) generation of cells. Inset; UTI89 grown in LB divides highly symmetrically. All values represent mean ± SD. Source data are provided as a Source data file.

size skew, however, seen at all generations, with ~80% of newborns shorter than average and ~10% of newborns at least twice the average length (Fig. 2k and Supplementary Movie SM6). We then tracked the size of each pinched-off cell for 3–4 of its subsequent divisions (see Fig. 2l), which revealed a slight decrease in mean length over four generations: 4.83 ± 2.33 μm ($n = 129$), 4.01 ± 1.71 μm ($n = 203$), 3.44 ± 0.67 μm ($n = 397$) and 3.11 ± 0.5 μm ($n = 664$), respectively (Fig. 2m). For comparison, cells that had not been through an infection cycle were on average 2.79 ± 0.35 μm ($n = 340$) in length at birth (Fig. 2m, 'No inf.'). The first and second divisions sometimes gave rise to abnormally long cells (Fig. 2m), but this corrected over the next one or two generations, with essentially all pairs of daughter cells in the fourth generation being the same length at birth (Fig. 2n).

**Nucleoid distribution during filament reversal**. To see how DNA partitioning was maintained during reversal, we also looked at the localization of chromosomes in filaments of UTI89

transformed with a plasmid expressing HupA-RFP[32] (which did not interfere with UTI89 growth; Supplementary Fig. 1). Fluorescence microscopy revealed that the DNA was distributed along the lengths of filaments and in somewhat irregular patches (Supplementary Fig. 4a, 0 min). Filaments contained on average 28.2 nucleoids per 100 μm cell length, corresponding to ~3.5 μm cell length per nucleoid, which is similar to normal rods. The average length of nucleoid masses in filaments without visible constrictions was 2.05 ± 0.86 μm ($n = 305$), with a primarily bimodal size distribution (Supplementary Fig. 4b, red). In post-divisional rods that had been pinched off from a filament, and were about to divide, the average nucleoid length was 1.3 ± 0.24 μm ($n = 254$) (Supplementary Fig. 4b, green), similar to non-infection UTI89 growing in LB of 1.45 ± 0.32 μm ($n = 308$) (Supplementary Fig. 4b, blue). There was an increasing degree of symmetry in chromosome partitioning over generations (Supplementary Fig. 4b, Inset), consistent with the observed progression of cell size symmetry (Fig. 2k).

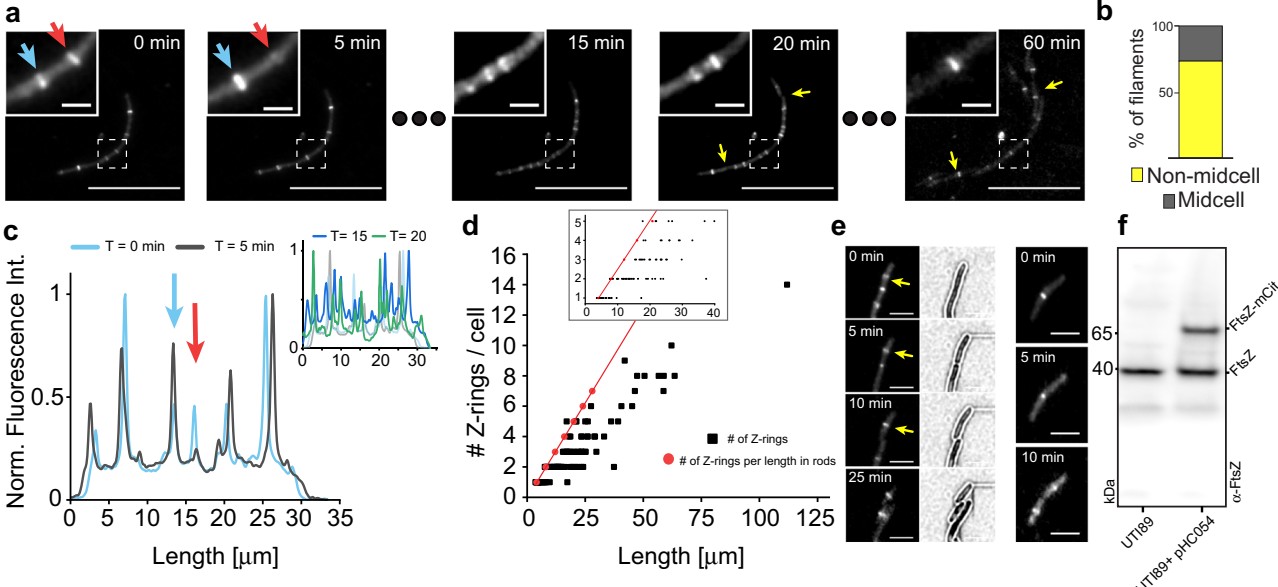

**Fig. 3 FtsZ-mCitrine localization and dynamics in filaments.** To visualize Z-rings in filaments, UTI89 cells were transformed with a plasmid producing FtsZ-mCitrine (pHC054) and run through the UTI model. **a** Still images from a representative time-lapse of a filament undergoing reversal (Supplementary movie SM7). Inset show a magnified view of the dashed square (Supplementary movie SM8), blue arrow indicates a FtsZ-mCitrine ring that increase in intensity from $t = 0$ to $t = 5$, red arrow indicates a FtsZ-mCitrine ring that completely disassembles during the same time. Yellow arrows at $t = 20$ indicate the first divisions. Scale bar 20 μm. **b** Percentage of filaments in which first division was at midcell vs non-midcell (towards the poles). **c** Normalized fluorescence intensity traces at various time points from the filament in **a**. Blue and red arrows correspond to the blue and red arrows in **a**. Inset show traces at $t = 15$ (dark blue) and $t = 20$ min (green), light blue and grey are the traces from earlier timepoints. **d** Filaments have less Z-rings assembled than the corresponding rods would have per cell length. Black squares show the number of FtsZ-mCitrine rings in filaments at the start of the imaging. Red dots indicate the corresponding number of 'expected' Z-rings assuming one ring per WT cell length (i.e., one ring per ~4 μm). Inset shows magnification of filaments in lengths up to 40 μm. $n = 100$. **e** The presence of rod-shaped cells after infection indicated that FtsZ-mCitrine did not influence division (nor induced filamentation). Left column: FtsZ-mCitrine was pre-assembled in 'cells' before they separated from the mother filament. Scale bar 4 μm. **f** Western blot indicating the FtsZ protein levels in filaments. The total production of the FtsZ-mCitrine was 43 ± 1% of the total cellular FtsZ pool ($n = 3$). All times in time lapses indicate time relative to first image ($t = 0$ min). Source data are provided as a Source data file.

**Dynamic repositioning of FtsZ-rings in IRF.** Our results above showed that filaments surviving the infections rapidly coordinate reversion to rods when conditions become favourable for vegetative growth. It is not known when the cell division machinery assembles in UPEC filaments to reinitiate division. To determine whether or not the divisome was at least partially pre-assembled in filaments, we transformed the UTI89 strain with a plasmid producing FtsZ-mCitrine[33] as a divisome marker, since all other essential division proteins are dependent on FtsZ assembly at midcell during vegetative growth[14,34]. Fluorescence microscopy of UPEC filaments expressing FtsZ-mCitrine released from the bladder cells after an infection showed that all filaments, regardless of length, had at least one FtsZ-ring assembled, but often more. Interestingly, however, movies revealed that FtsZ-rings formed, disassociated, and re-formed multiple times in various locations within the same filament (Fig. 3a, c and Supplementary Movies SM7 and SM8), and did not always end in a division event. There were on average less FtsZ rings per unit length in filaments compared to rods (i.e., one ring per ~4 μm) (Fig. 3d). These results indicate that FtsZ rings are significantly less stable in filaments than in rods. The dynamic repositioning of FtsZ rings suggests that in IRF the divisome is regulated at the stage of Z-ring stabilization or maturation.

The presence of FtsZ may help to explain how the first newborn daughter cells from filaments were able to quickly divide again (~15 min, Fig. 2h). Consistent with our results for wild-type UTI89, when the first divisions took place in filaments labelled with FtsZ-mCitrine, the majority (~75%) were away from midcell and predominantly towards the poles (Fig. 3a, b, yellow arrows,

and Supplementary Movie SM7). Some FtsZ-rings were also inherited by daughter cells after division had occurred elsewhere (Fig. 3e, left column, yellow arrows).

**Knockout of *damX* affects vegetative cell division and IRF in UTI89.** Since FtsZ rings did not always result in a division even, it is likely that other division regulators are involved. We turned our attention to DamX, as it was previously found to be essential for IRF in UTI89[28], though how it regulates IRF is unknown.

Previous studies on DamX showed it is localized to the divisome during vegetative growth in a strictly FtsZ-dependent manner[35], but its exact role in division has been uncertain since *damX* deletion causes no obvious division defect in *E. coli* K-12[22,23] (Fig. 4a, b). We found that UTI89Δ*damX* grew at rates indistinguishable from WT in midlog LB cultures. However, the cells were on average ~33% longer than WT (UTI89Δ*damX* = 5.25 ± 3.38 μm ($n = 228$), and WT UTI89 = 3.94 ± 1.14 ($n = 171$)) (Fig. 4c, d and Supplementary Fig. 5). Most noticeably, UTI89Δ*damX* showed a skewed distribution of cell size compared to WT; ~15% of UTI89Δ*damX* cells were 8 μm or longer, which is at least twice the average WT UTI89 length (Fig. 4d, Supplementary Movie SM9, and Supplementary Fig. 5). UTI89Δ*damX* cells producing mEos3.2-DamX from a plasmid (pMP6) showed a mean cell length of 4.87 ± 1.41 μm ($n = 219$) and a substantial rescue of the elongated population, with ~3% cells longer than 8 μm, compared to more than 15% for the uncomplemented UTI89Δ*damX*, and around 1% for WT UTI89 (Fig. 4e, f and Supplementary Fig. 5). Western blots indicated that DamX

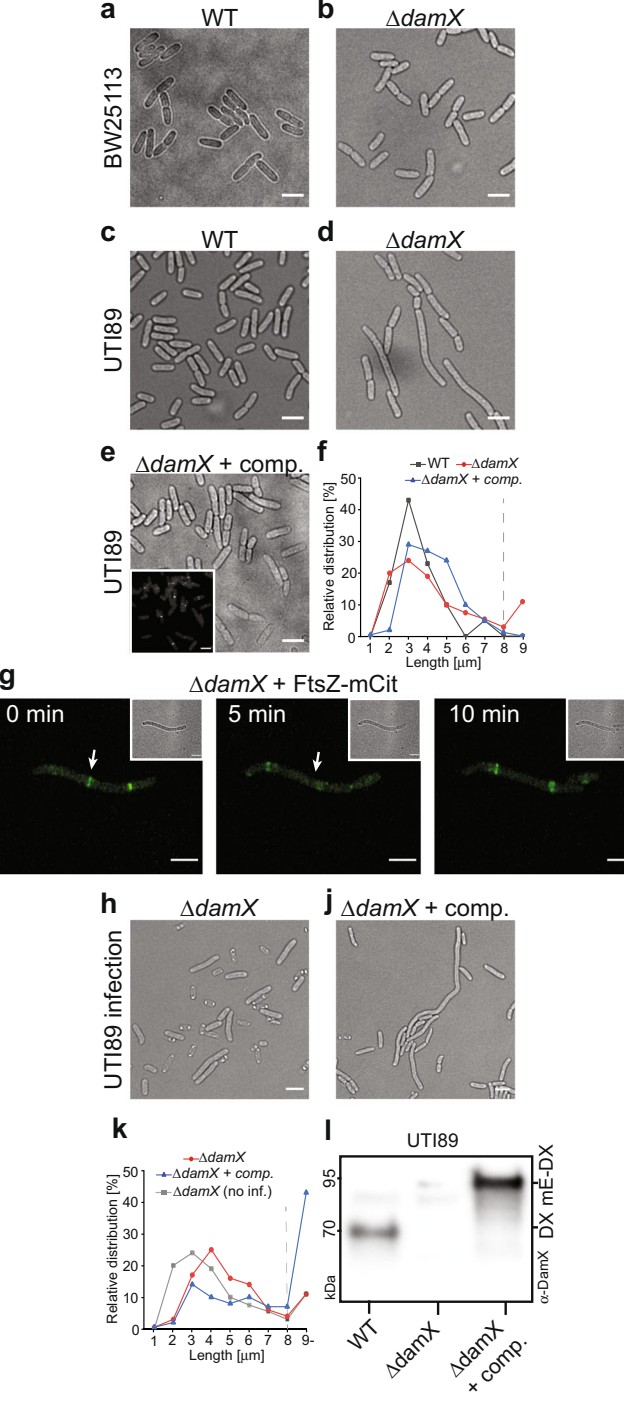

**Fig. 4 UTI89Δ*damX* displays an elongation phenotype in rich medium but does not produce filaments in a model UTI.** Deletion of *damX* in a non-pathogenic *E. coli* (strain BW25113) does not give rise to a phenotype. **a** Representative image of WT BW25113 cells **b** Representative image of BW25113Δ*damX* cells. **c** Representative image of WT UTI89 cells. **d** Around 15% of cells in UTI89Δ*damX* displayed a division defect resulting in abnormally long cells. The overall range of cell lengths for UTI89Δ*damX* was 25 μm, with >90% of the long cells being in the range 7–15 μm. We did not observe any clear correlation between length at birth and abnormal divisions (Supplementary Movie SM9). **e** The long phenotype could largely be reversed when complementing DamX with a plasmid producing mEos3.2-DamX (pMP6). Inset shows fluorescence localization of mEos3.2-DamX in the pre-converted green channel. **f** Relative distribution of cell lengths of the UTI89 strains grown in rich media. The number on *x*-axis represents bins of one μm, except for last bin which encompasses all cells longer than 9 μm. Grey dashed line indicate 8 μm in length. **g** Representative elongated UTI89Δ*damX* cell. Initially are two Z-rings (FtsZ-mCitrine) assembled along the cell body. White arrow shows one of the Z-rings disassemble without a division event. Inset show corresponding bright field image. **h** UTI89Δ*damX* do not filament in the UTI model. This is consistent with what has previously been observed[28]. **j** UTI89Δ*damX* complemented with mEos3.2-DamX produced from a plasmid does filament. This suggests that the mEos3.2-DamX is a functional protein fusion. **k** Length distribution of UTI89Δ*damX* (red) and UTI89Δ*damX* + complementation (blue) cell from infections. UTI89Δ*damX* cells from LB growth are included for reference (grey). **l** DamX levels in various UTI89 strains after a round of infection. Δ*damX* indicate UTI89Δ*damX*, Δ*damX* + comp indicate UTI89Δ*damX* complimented with pMP6. DX = DamX, mE-DX = mEos3.2-DamX (pMP6). Note that both DamX and mEos3.2-DamX ran at a higher molecular weight than expected, we do not know the cause at this time but is similar as to what has previously been seen for DamX by others[28]. All scale bars = 4 μm. Source data are provided as a Source data file.

In the infection model, we observed that UTI89Δ*damX* failed to undergo IRF, consistent with the previous study[28] (Fig. 4h). Instead, we saw a moderate level of elongation in a subpopulation of cells, as seen in the LB cultures. Interestingly, we also saw a relatively high proportion (~10%) of short cells that appeared translucent indicating a loss of cell integrity (Fig. 4h). We speculate that cells may be attempting to filament but were unable due to the deletion of *damX*. Use of the UTI89Δ*damX*/pMP6 strain largely restored IRF and cell integrity, indicating that the mEos3.2-DamX protein fusion is mostly functional and confirms that DamX is responsible for the observed phenotypes (Fig. 4j and Supplementary Movie SM11). The mean length was 13.17 ± 15.72 μm (*n* = 184), with more than ~50% of the cells 8 μm or longer (Fig. 4k, Supplementary Fig. 5). DamX production from the plasmid was ~50% higher than WT (Fig. 4l) and did not interfere with growth or division of filaments (Supplementary Movie SM11). We conclude that DamX influences filamentation and cell integrity during infection.

**A switch in DamX localization is associated with the onset of filament division.** To investigate how DamX acts in IRF and filament division, we visualized mEos3.2-DamX produced from an anhydrotetracycline-inducible expression plasmid (pDD7) in UTI89. [Note, for unknown reasons mEos3.2 fluorescence was too weak in the constitutive expression plasmid pMP6.] In UTI89/pDD7, mEos3.2-DamX fusion represented ~60% of the total cellular DamX (Supplementary Fig. 1c), which did not affect vegetative growth or division (Supplementary Fig. 1a, b), consistent with previous studies[36–38]. High-level DamX overexpression has previously been shown to cause filamentation[39,40], and its strong upregulation is

production levels were similar in *E. coli* K-12 (BW25113), UTI89 and UTI89Δ*damX* + pMP6 (Supplementary Fig. 5d). Thus, loss of DamX results in a moderate vegetative division defect in UTI89, which is not seen in the model K-12 strain.

We then produced FtsZ-mCitrine to a level that does not interfere with growth or division[36] (Supplementary Fig. 5e) in UTI89Δ*damX*. In midlog LB cultures, FtsZ-mCitrine localized to midcell in the normal-sized rods, followed by constriction and division, like in the WT. However, in the elongated cells, the FtsZ rings frequently formed and then disassembled without division (Fig. 4g, white arrows, Supplementary Movie SM10), reminiscent of FtsZ-mCitrine in IRF (Fig. 3). DamX thus appears to influence FtsZ ring stability in UTI89 filaments.

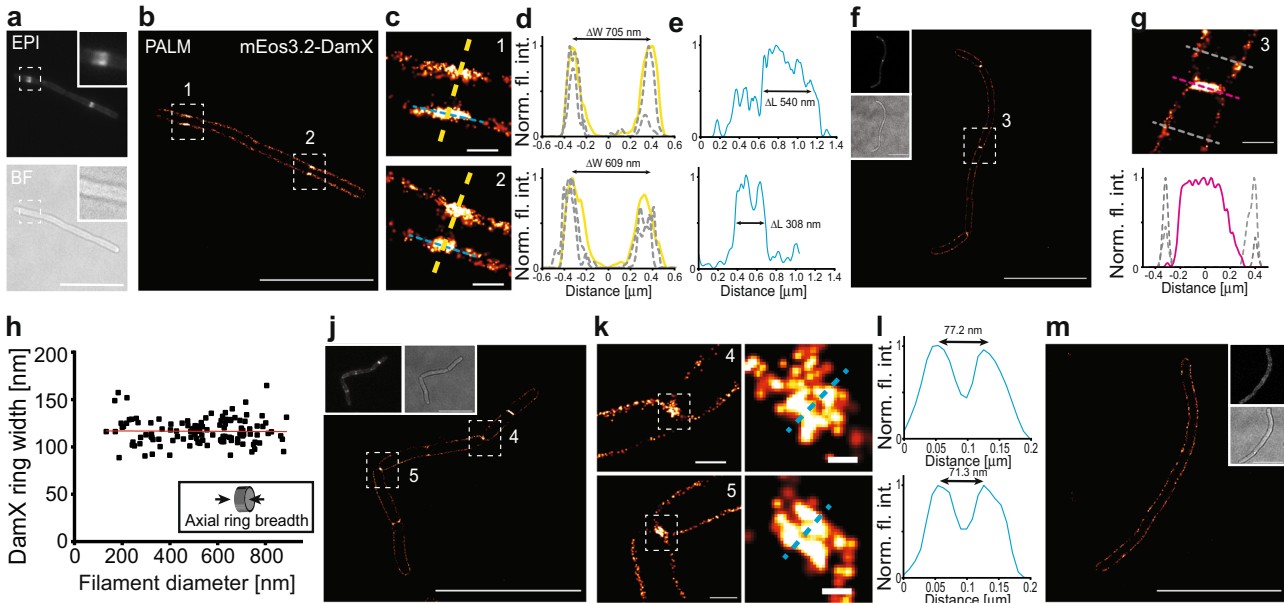

**Fig. 5 DamX localizes at the division site prior to membrane constriction.** Filaments expressing mEos3.2-DamX harvested from the back-end of flow chambers after infections were grown in LB for 1 h before imaged using single-molecule microscopy. **a** mEos3.2-DamX localized at multiple division sites simultaneously and accumulated prior to visible invagination of the membranes. Insets: Close-up images of mEos3.2-DamX and the corresponding bright-field image. **b** PhotoActivatable Localization Microscopy (PALM) image of the same filament as in **a**. **c** Close-up images (1) and (2) of the mEos3.2-DamX ring assemblies at division sites prior to condensing into a ring structure (from **b**). **d** Fluorescence intensity plots of the mEos3.2-DamX assemblies and membrane widths. Plots showing ring and membrane widths; yellow lines represent ring assemblies; grey dotted lines represent membrane widths 1 μm up and downstream of the mEos3.2-DamX accumulation. ΔW indicates the peak-to-peak distances. **e** Length of the molecule assemblies along the length axis of images in **c**. ΔL indicates the length of the intensity profile at 50% of the intensity. **f** PALM image of a typical filament during reversal. **g** Close-up image of a constricting mEos3.2-DamX ring (3), fluorescence profile underneath: yellow line represents the ring, grey represents membrane width. **h** Axial breadth of mEso3.2-DamX rings along the length of the filaments at various cell diameters. Average width 116.5 ± 13.4 (n = 122), values represent mean ± SD. Red line represents linear fit to the data: $y = -0.009*x + 116$. **j** mEos3.2-DamX remains at the old division septum after membrane separation. **k** Close ups of old division sites from **j** (4) and (5). **l** Peak-to-peak distance of fluorescence intensities of the membrane assemblies of mEos3.2-DamX. **m** A typical filament before division sites have been defined. Scale bars (**a**, **b**, **f**, **j**, **m**) = 10 μm, (**c**, **g**, **k**$_{left}$) = 500 nm, (**k**$_{right}$) = 100 nm. Source data are provided as a Source data file.

associated with the UTI89 filamentous response during infection[28]. We found that the moderate production of mEos3.2-DamX in UTI89 did not inhibit filament reversal (Figs. 4 and 5 and Supplementary Figs. 1c and 5). Furthermore, in bacteria released from infections, mEos3.2-DamX localization and dynamics in the rod-cell population were similar to what has been seen in K-12 during vegetative growth[22] (Supplementary Movie SM12).

To determine the localization and subcellular structures of mEos3.2-DamX in UPEC filaments, we imaged UTI89/pDD7 obtained from infections using live-cell single-molecule Photo-Activated Localization Microscopy (PALM). Filaments at various stages of preparation for division were identified in these samples. Those that appeared to be at an early stage of preparing to divide were identified by the appearance of broad mEos3.2-DamX rings prior to visible invagination (Fig. 5a, b). Consistent with this, measurement of the diameter of the broad mEos3.2-DamX rings and of the filament 1 μm either side showed a ratio of 1:1.05–0.95 (Fig. 5c, d). The fluorescence intensity profiles along the filament axis showed that the pre-constriction mEos3.2-DamX localizations were as broad as >500 nm (Fig. 5c, e). Active division was identified by a significantly narrower diameter of the mEos3.2-DamX ring than the filament (Fig. 5f, g). The mean axial breadth of these condensed mEos3.2-DamX rings along the cell length was 116.5 ± 13.4 nm (n = 122) (Fig. 5h), similar to midlog LB samples (102.5 ± 20.2 nm, n = 150) (Supplementary Fig. 6). In accordance with previous observations of another SPOR-domain protein, FtsN[41], mEos3.2-DamX seemed to linger at the division

sites after the inner membranes had closed, before being completely disassembled (Fig. 5j–l). Interestingly, live pre-divisional filaments were also identified, in which mEos3.2-DamX showed dispersed cell envelope localization with no clear rings (Fig. 5m). This suggested that filament formation may not require ring localization of DamX.

Given that DamX is essential for IRF, we sought to capture its behaviour during the switch from filament elongation through to completion of division and several subsequent generations. Time-lapse epifluorescence imaging of UTI89/pDD7 filaments collected from infection and recovering on LB agar pads revealed mEos3.2-DamX assembled as rings at different places along the length of filaments (Supplementary Movie SM13). Figure 6a shows the formation of at least three generations of mEos3.2-DamX division rings in one filament over time, and importantly, that the filaments then reverted into rods at those sites. In contrast to what we observed for the FtsZ-mCitrine rings, once an mEos3.2-DamX ring formed, it remained in place until division was completed.

Measurements of mEos3.2-DamX ring position indicated that there was a preference for rings to form initially at the approximate ¼, ½ or ¾ positions along the filament length regardless of the length, which was predominantly in the range of ~8–40 μm (n = 134) (Fig. 6b). The data also indicated that cells pinching off in the second and third generation did so closer to the poles (Fig. 6b, c). Note that the mother filament would have grown in length between each division.

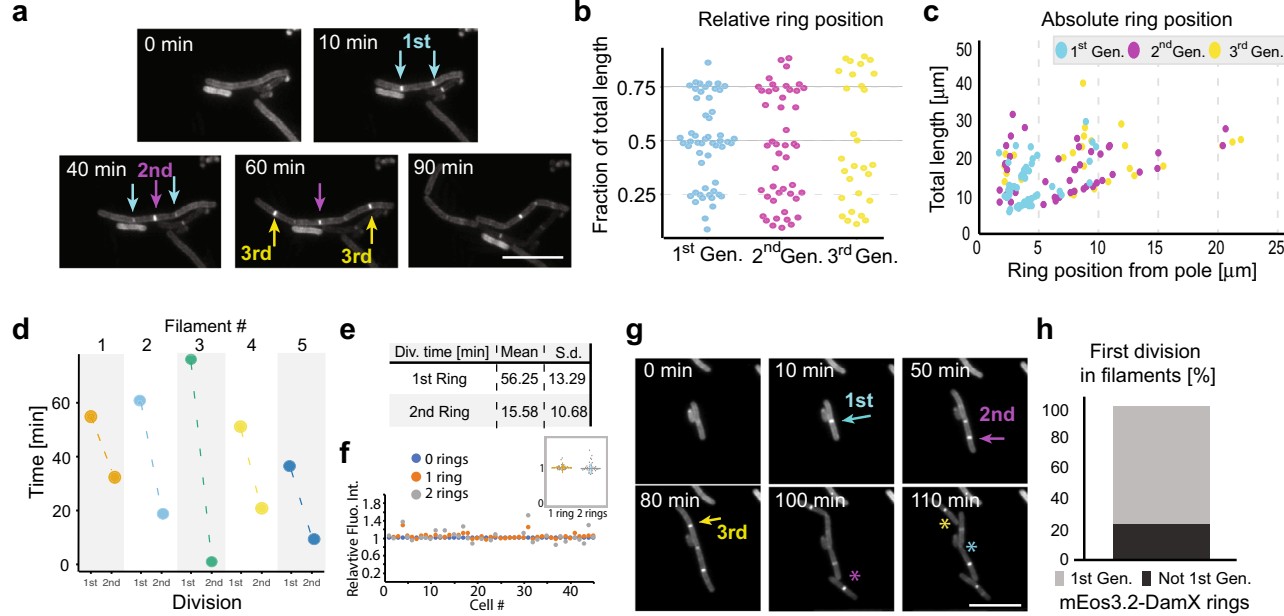

**Fig. 6 mEos3.2-DamX localization and dynamics in filaments. a** Time-lapse images of filaments expressing mEos3.2-DamX (Supplementary movie SM14). Formation of mEos3.2-DamX rings was followed over time in filaments where no rings were observed in the first image. Cyan arrows show first generation, magenta arrow second generation and finally yellow arrows indicate the third generation division rings. **b** Relative de novo positioning of mEos3.2-DamX division rings in filaments. Most rings assembled at locations close to 1/4, 1/2 or 3/4 of the total filament length. Colour coding follows that in **a**. Gen. = Generation. $n = 134$. **c** The distance of mEos3.2-DamX rings in μm from one of the cell poles. **d** Plot shows the time from first mEos3.2-DamX ring formation to the first division (1st), and the time from the first division to the second (2nd, $\Delta t = t_2 - t_1$) for five randomly picked filaments. **e** Summary showing average times of first and second division based on mEos3.2-DamX ring formation and constriction. **f** Total cellular mEos3.2-DamX fluorescence intensity did not change with formation of division rings. Blue dots represent total integrated fluorescence normalized to filament area one frame before formation of the first ring, orange dots represent integrated fluorescence when one ring had formed, grey dots represent integrated fluorescence when two rings had formed. **f** inset, Relative mean cellular fluorescence with one ring = $1.02 \pm 0.07$ ($n = 45$), relative mean cellular fluorescence with two rings = $1.03 \pm 0.15$ ($n = 32$). Values represent mean ± SD. **g** A shorter filament from a time-lapse movie indicating that the second generation of mEos3.2-DamX ring (magenta, formed at $t = 50$) constricted and pinched off prior to the first ring formed (cyan, formed at $t = 10$ min). First image showing a full division is at $t = 100$ min, at the place where the second mEos3.2-DamX ring was formed (indicated by a magenta asterisk). Division of the first mEos3.2-DamX ring is indicated by cyan asterisk. A third-generation mEos3.2-DamX ring is indicated by the yellow arrow (with division indicated by yellow asterisk). Scale bar 10 μm. **h** In all, 23% of the second generation mEos3.2-DamX rings formed in filaments pinched off before the first generation mEos3.2-DamX rings. All scale bars = 10 μm. Source data are provided as a Source data file.

In line with our observations that the apparent time between division of daughter cells from filaments decreased with successive generations, we wanted to know if this was also consistent in the first division (previously, we could only determine the relative time from the second division without a division marker, i.e., Fig. 2). That is, would the first division ring be present longer than the second before a division? To do this, we followed mEos3.2-DamX ring formation, constriction, and subsequent division in filaments over time. Indeed, the average time from the first mEos3.2-DamX ring formation to the first division was $56 \pm 13$ min (SD), and there was a significant reduction in time to the next (second) division: $15 \pm 10$ min (SD) (Fig. 6d, e). This was also in good agreement with our observation for the FtsZ-mCitrine rings that completed constriction (Fig. 3e).

We also wanted to see if the intensity of mEos3.2-DamX varied during ring formation, as this would be proportional to the total cellular expression level. The total cellular fluorescence did not change significantly with the formation of new mEos3.2-DamX rings (Fig. 6f); compared to filaments with no visible rings, the relative mean cellular fluorescence with one ring was $1.02 \pm 0.07$ ($n = 45$), while relative mean cellular fluorescence with two rings was $1.03 \pm 0.15$ ($n = 32$) (Fig. 6f, inset). Curiously, we also noticed that the mEos3.2-DamX rings did not always divide in the order that they initially formed (Fig. 6g). It was observed that around

20% of filaments had the second generation of rings pinch off prior to the first (Fig. 6h).

Since mEos3.2-DamX was assembled at multiple sites along filaments, we wondered whether septal PG (sPG[42]) synthesis/hydrolysis also occurred at multiple sites at the same time. To test this, we pulse-labelled filaments during reversal to detect regions of sPG synthesis/hydrolysis with an OregonGreen488-labelled Fluorescent D-amino acid (OGDA)[43,44]. As we saw with mEos3.2-DamX, OGDA was observed in multiple fluorescent rings (Fig. 7a), indicating that active peptidoglycan synthesis was present at multiple sites and that multiple cell division machineries were active in multiple rings simultaneously. We also observed various levels of OGDA intensity, indicating differing sPG activity in the rings. The localization of OGDA at many division sites preceded visible constriction, suggesting that the cell division machineries would be assembled at that stage, whereas other OGDA localizations were clearly associated with later-stage division constrictions in both filaments and rods (Fig. 7b, c).

## Discussion

We have used live-cell time-lapse, epifluorescence and super-resolution microscopy to quantitatively characterize the growth and division of filamentous Uropathogenic *E. coli* (UPEC)

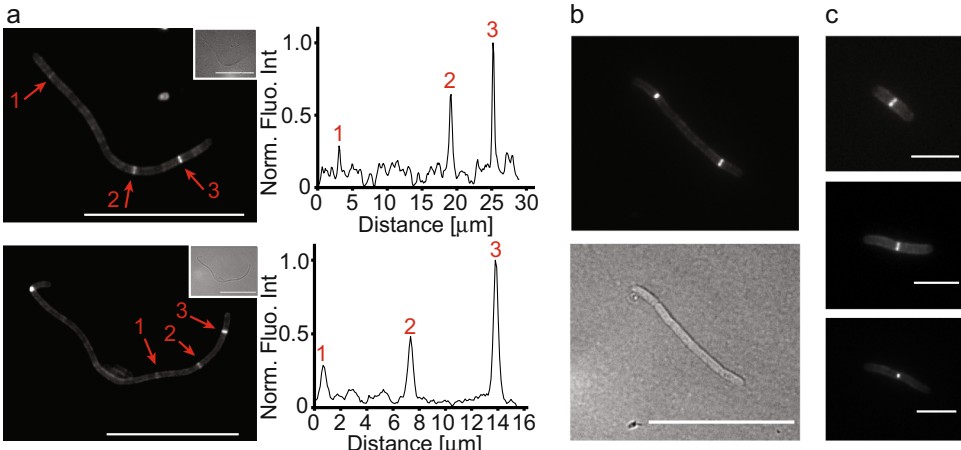

**Fig. 7 Peptidoglycan synthesis is active at multiple locations at the same time.** WT UTI89 filaments from an infection cycle were labelled with the green fluorescent D-amino acid (FDAA) probe OGDA and imaged using fluorescence microscopy. Harvested filaments were pelleted and resuspended in LB and placed at 37 °C for 2 h before OGDA labelling for 5 min. **a** Multiple active sites for peptidoglycan synthesis can be observed in filaments. Inset: corresponding brightfield images. Fluorescence intensity plots for selected sections are shown next to the images (peaks numbered). **b** A representative mid-length filament with two clear OGDA accumulations at division sites. **c** Rod-shaped cells in various stages of membrane constriction. Cells had already reverted from filaments and were then stained with OGDA, showing strong fluorescence accumulation only at midcell, as expected. Scale bars **a**, **b** = 20 μm, **c** = 4 μm. Source data are provided as a Source data file.

released from the dispersal stage of infection of bladder epithelial cells[29,30]. UPEC filaments can grow to hundreds of microns long in UTI, but we found that shorter filaments (<50 μm) were more likely to survive and revert to rods in culture. UPEC filamentation has been proposed as a survival strategy to prevent their potential phagocytosis by immune cells or improve surface adhesion[10,11,13]. Our results suggest that UPEC filamentation is a response to stress during dispersal and exposure to human urine —and the stress is not always overcome.

Prior to filament reversal (i.e., division into rods) most filaments undertook a period of variable growth (on LB agarose), where elongation rates ranged from 0.08 μm min$^{-1}$ to 1.72 μm min$^{-1}$, or up to ~20 times the elongation rate of a standard rod cell. Since filamentation combines the growth power of multiple rods and effectively links the strength of their cell walls together in one cell body, we propose that the rapid directional elongation of filaments physically aids UPEC dispersal from the IBC biofilm and host cell carcass, helping them to 'push out' into the extracellular environment. Indeed, the extrusion-like images of UPEC filaments exiting host cells are consistent with this potential selective advantage[29,45].

The elongation rate of UPEC filaments was not correlated with the initial length of the filament, indicating a substantial variability in metabolic states of filaments regardless of length. Consistent with this, the time to the first division ranged from just a few minutes to more than two hours. The substantial heterogeneity may be related to both the asynchronous dispersal from the bladder cells and the different morphological fates of individual IBC dispersal events[11,13,29].

Most UPEC filaments divided near the ends, releasing a smaller 'pinched-off' cell. Over the first few divisions, the filament interdivision time became shorter, much like filament reversal after antibiotic treatment[31]. Furthermore, the pinched-off UPEC cells became shorter over the initial generations too, approaching the size of standard rods. The combined results suggest that division tends towards balanced growth over a few generations. Previous studies looking at the cell division protein FtsA during reversal of *E. coli* filaments induced by antibiotics and DNA damage (SOS response) found that division rings assemble evenly along the cell, sometimes only transiently and then at new locations, with one division event at a time occurring at the time[31]. In

the current study, multiple overlapping divisions were common during reversal of IRF, predominantly towards the poles. Despite the strain and condition differences between the studies, the findings suggest that significantly different mechanisms of filamentation and reversal underpin the antibiotic, SOS, and IRF responses, highlighting the importance of understanding these differences at both strain and protein function levels for future development of infection therapeutics.

We investigated the role of two distinct cell division proteins FtsZ and DamX in UTI89 filaments. FtsZ has long been regarded as the main initiator of the division and assembly of its associated machinery in most bacteria. DamX, a member of the SPOR-domain family of proteins, has a less well-defined role but is known to localize to midcell in an FtsZ-dependent manner during vegetative growth. In cephalexin-induced filaments, FtsZ rings predominantly form at constant intervals[46]. To test if this was the case also in IRF and take a step towards understanding the mechanism of division arrest in IRF, we visualized how fluorescently tagged FtsZ behaved in UTI filaments. In contrast to what was observed with cephalexin, the Z-rings in UTI filaments were formed and spaced in highly irregular intervals. On the other hand, similar to what was seen for the antibiotic-induced filaments[31], Z-rings were unstable in IRF and showed dynamic repositioning prior to division. This suggests that FtsZ ring instability is a part of the division arrest mechanism leading to filamentation, and that the eventual re-initiation of division in IRF is regulated by another protein that controls the behaviour of FtsZ during the switch from filament elongation to division. Once a FtsZ ring started to constrict, it remained at the division site until a late stage when it disassembled similar to what has been observed in rods[37].

A recent study found that DamX—previously implicated in sPG remodelling during vegetative growth—was upregulated during the dispersal stage of infection when filamentation occurs and is also essential for UPEC filamentation[28]. This led to the hypothesis that DamX switches functions to block division at potential division sites during UPEC filamentation. Surprisingly, we instead observed that fluorescently-tagged DamX (mEos3.2-DamX) was initially not assembled at potential division sites in UPEC filaments, but was dissipated around the cell envelope and

assembled later as rings where filaments eventually divided. In contrast to what we observed for FtsZ, once a DamX ring was formed it remained formed, a division event always followed, and the DamX rings were stable until the completion of division. Our single-molecule data suggest that mEos3.2-DamX assembles into rather broad rings at the quarter or half positions along filaments, consistent with the expected locations of division, and then condenses leading up to constriction, where mEos3.2-DamX rings showed very similar dimensions as those formed by related cell division components during *E. coli* division[36,47,48]. Lastly, mEos3.2-DamX remained assembled at the division septum until after the inner membranes were visibly separated. The lingering of the protein at the division site suggests a role for DamX in sPG regulation until building of the new cell wall is completed.

In summary, the current data suggest that during filamentation, DamX has a role in blocking division that involves dissipated localization around the whole membrane, possibly associated with sequestration of division proteins or the direct inhibition of division throughout the length of the filament. DamX then switches function for reversal to assist in cell division in a manner that structurally resembles normal cell division. DamX also has an apparently more significant role in vegetative division of UTI89 than in the traditional model strain, K-12. DamX has also been implicated in morphology changes and virulence in other pathogens (e.g., *Vibrio parahaemolyticus*), indicating a broader overall function of the protein across species[49]. Further studies on diverse strains will tell if one is atypical, but the difference may also reflect the apparent importance of DamX to the UPEC infection cycle. We envision that the IRF response and the infection model may become an increasingly important tool for the study of morphology, division and survival of pathogenic bacteria.

## Methods

**Ethics approval.** This study has human research ethics approval from the UTS Human Research Ethics Committee (HRCH REF No. 2014000452). All urine donors gave their approval to participate in this study by informed consent under this study's ethics approval number above.

**Bacterial cell growth.** A single colony of respective *E. coli* UTI89 strain was grown overnight in a 20 ml LB (Difco #244620) culture at 37 °C without shaking, to favour the expression of type-1 pili, which facilitates adhesion to the bladder cells during infection[5]. Antibiotics were added when needed (ampicillin 100 μg ml$^{-1}$ (Sigma #A9518), spectinomycin 25 μg ml$^{-1}$ (Sigma #S4014)). The following morning, the cultures were pelleted and resuspended in 1× PBS (Bio-Rad #1610780) to a concentration of OD$_{600}$ 0.2 and added to the infection model. Cells not used in the infection model were diluted 1:50 and grown to an OD$_{600}$ of ~0.4 before imaging to measure cell length and fluorescence profiles. Bacterial strains and plasmids used in this study are listed in Table 1.

**Plasmid construction.** The expression plasmid pAJM.011 was used as a backbone in the construction of pDD7 (mEos3.2-DamX). This plasmid was part of the Marionette sensor collection[50], which was obtained from Addgene (Kit #1000000137). Initially, the *aph* coding sequence in pAJM.011 was replaced by the *bla* coding sequence. In a second step, the coding sequence for EYFP was replaced by the coding sequences for mEos3.2 and FtsN. Finally, the coding sequence for FtsN was replaced by that for DamX. All fragments were amplified by polymerase chain reaction (PCR) using Q5 DNA polymerase (New England Biolabs, USA, NEB #M0491). Fragments to be ligated contained regions of 20-30 bp of homology, and were cloned using the in vivo DNA assembly method[51]. The coding sequence for *bla* was obtained from pET15b, mEos3.2 from Genscript (The Netherlands) and FtsN/DamX from the *E. coli* strain MG1655. Oligonucleotide synthesis and DNA sequencing was performed by Eurofins Genomics (Germany).

pGI4[29] was used as a plasmid backbone for the construction of a plasmid constituently producing mEos3.2-DamX. Firstly, pGI4 was digested with NcoI and BamHI-HF (New England Biolabs, #R0193S and # R31136R, respectively). mEos3.2-DamX was amplified from pDD7 using Q5 DNA polymerase (New England Biolabs) with oligonucleotides synthesized by Integrated DNA Technologies (USA). The products of the previous reactions were cloned using Gibson[52] assembly and used to transform DH5α. The resulting plasmid, denoted pMP6, was purified and sequenced by AGRF (Australia). All primers used in this study are available in the Supplementary Information file.

**Table 1 Table of figures and sources of bacterial strains and plasmids.**

| Figure | Strain | Plasmid | Fluorescent protein/dye | Source |
|---|---|---|---|---|
| Fig. 1 | UTI89 | pIG5 | sfGFP | 29 |
| Fig. 2 | UTI89 | pIG5 | sfGFP | 7 |
| Fig. 3 | UTI89 | pHC054 | FtsZ-mCitrine | 33 |
| Fig. 4 | UTI89 UTI89Δ*damX* BW25113 BW25113Δ*damX* | pMP6 pHC054 | mEos3.2-DamX FtsZ-mCitrine | This study and refs. 22, 28, 33 |
| Fig. 5 | UTI89 | pDD7 | mEos3.2-DamX | This study |
| Fig. 6 | UTI89 | pDD7 | mEos3.2-DamX | This study |
| Fig. 7 | UTI89 | --- | OGDA | 43 |

Italics indicate gene name.

**Fluorescent protein production.** WT UTI89 strain was transformed with pGI5[29], pMP6 or pDD7 to produce strains expressing sfGFP or mEos3.2-DamX, respectively. sfGFP (from pGI5) and mEos3.2-DamX (from pMP6) did not need inducer as it was produced from a constitutive promoter[29]. mEos3.2-DamX (pDD7) expression was induced by adding 1 μM anhydrotetracycline (aTc) for 1 h to the infection cultures prior to collecting filaments.

FtsZ-mCitrine (pHC054)[33] was induced by adding 5 μM isopropyl β-D-1-thiogalactopyranoside IPTG for 1 h to the infection cultures prior to collecting filaments. We confirmed that the expression level of the FtsZ-mCitrine construct was not interfering with growth and division in neither rods nor filaments (Fig. 3e and Supplementary Fig. 1). Previous studies have shown that production of fluorescent protein fusions to FtsZ (e.g., FtsZ-mEos3 and FtsZ-GFP) below ~ 70% of total cellular FtsZ does not measurably alter cellular growth, divisions nor Z-ring morphology[36]. FtsZ-mCitrine in the UPEC filaments was produced to less than 50% of total cellular FtsZ (Fig. 3f).

WT UTI89 strain was transformed with pSTC011[32] to produce a strain expressing HupA-RFP. HupA-RFP was induced for 1 h by adding 20 μM IPTG to the cultures containing filaments.

**Preparation of human urine samples.** Urine from two different donors (one female and one male) was collected in the morning and stored for at least 2 days at 4 °C. Samples were centrifuged at 4500 × *g* for 8 min and the supernatant was filtered through a 0.2 μm membrane filter. The specific gravity was determined by comparison to pure water using a gravity meter. Only urine samples in the pH range 5.12-5.83 and USG range 1.024–1.031 g ml$^{-1}$ were used[29]. Filter sterilized samples were frozen at −20 °C until required. Note, to be as consistent as possible, only one batch of urine was used per technical replicates, since both pH and USG can alter the proportion of filamentation in the samples[13].

**Urinary tract infection model.** We based our infection model on a previously described approach[13,29], with minor modifications. In summary, on day 1, flow chambers (IBIDI μ-Slides I$^{0.2}$ Luer, Cat#: 80166) were seeded with PD07i[53] epithelial bladder cells at a concentration of ~3 × 10$^6$ cells in EpiLife Medium (Gibco, #MEPI500CA) supplemented with growth supplements and antibiotics (HKGS, #S0015, and 100 μg ml$^{-1}$ Pen/Strep, SIGMA #P4333). The channels were left overnight for cells to adhere and grow into a confluent layer. The next day, flow channels were connected to New Era pumps via tubing and 20 ml disposable syringes, and a flow (15 μl min$^{-1}$) of fresh EpiLife (supplemented with HKGS) without antibiotics was maintained for 18–20 h. On day three, to induce infection, bladder cells were exposed to bacterial cultures at a concentration of OD$_{600}$ 0.2 for 20 min at a flow rate of 15 μl min$^{-1}$. Following this step, the media was changed back to EpiLife (supplemented with HKGS), after an initial flow of 100 μl min$^{-1}$ to flush out the excess bacteria, and flowed for 9 h to allow bacteria to adhere to and invade the epithelial bladder cells. This step was followed by flow (15 μl min$^{-1}$) of EpiLife (supplemented with HKGS) in the presence of 100 μg/ml gentamycin for 20 h to allow for the formation of intracellular bacterial communities (IBCs), as well as to kill and wash away any lingering extracellular bacteria. Following this 20 h, the media was changed to human urine (with pH between 5.12 and 5.83 and Urine Specific Gravity of at least 1.024 g ml$^{-1}$[29]) with flow (15 μl min$^{-1}$) for at least 20 h to induce filamentation and dispersal of the bacteria from the bladder cells. Since the proportion of live filaments is expected to vary with different batches and sources of urine[13]; to avoid variability we used urine from the same batch in experimental replicates. Filaments were collected from the back-opening of the

flow channels and resuspended in LB for fluorescent protein expression, *Live/Dead* cell staining and direct imaging, or stained (FDAA labelling) and fixed.

**Live/Dead cell staining**. Staining was performed as recommended by the manufacturer (LIVE/DEAD *BacLight* Bacterial Viability Kit, #L7012, Molecular Probes). Briefly, 3 μl of each dye mixture (SYTO9 and Propidium iodide) was added for each mL sample culture and mixed thoroughly. The dye/sample mixture was incubated for 15 min in the dark at room temperature. After incubation, 2–3 μl culture was spotted directly onto agarose pads (1.5% w/v in LB) for imaging. We also confirmed that cells and filaments divided and reverted using only SYTO9 labelling (Supplementary Fig. 3).

**FDAA labelling**. WT UTI89 filaments were labelled with 1 mM OGDA[43], using a protocol similar to what has been described before[54], for 5 min and washed twice in PBS, before fixation in 70% ice-cool ethanol for 1 h. After ethanol fixation, cells were washed twice in PBS and placed on agarose pads (1.5% w/v in water) for imaging.

**Microscopy and imaging**. Samples were placed on pre-made agarose (1–1.5% w/w) pads in LB (for dynamics) or M9 (single-molecule imaging) media in 65 μl gene frames (Thermo Scientific, AB0577), left to immobilize and imaged directly thereafter.

Live cell epifluorescence and bright-field time-lapse imaging of filaments reverting back to rods was done using a Nikon Ti2-E deconvolution microscope with extra-large Field Of View optics (full FOW of the microscope was 25 mm, this ensured us that long filaments could always be imaged within one FOW), equipped with a CFI Plan Apo Lambda DM ×100 oil objective (NA 1.45), and an environmental chamber set at 37 °C (Okolab cage incubator). Images were acquired every 2–10 min as required. Excitation of fluorophores was performed using a Lumencore Spectra II module, and fluorescence was detected using a back-illuminated Andor Sona 4.2 sCMOS camera. GFP and mEos3.2 (in the green state) emission was collected through a FITC filter and RFP emission through a Cy5 filter. Filters were from Semrock.

Live cell PALM imaging was performed on a Nikon (Ti2-E) N-STROM v5 with NIS v.5.30 using a 100x 1.49 NA oil objective in TIRF mode. Cover glass slides were washed with 95% EtOH, air dried, cleaned for at least 3 min with a plasma cleaner (Harrick plasma, PDC-23G) and used within 15 min of cleaning. Imaging was consistently performed at room temperature (~23 °C). Drift correction during image acquisition was minimized using the integrated PFS4 (Perfect Focus System). 100 nm multi-colour TetraSpeck beads were used as fiducial markers. Prior to single-molecule acquisition, the green state of mEos3.2 was excited by a 488 nm laser to acquire an epifluorescence image using a FITC emission filter cube. For single-molecule acquisition, mEos3.2 was photoconverted to its red state continuously by a 405 nm laser with increasing working powers ranging between 0.1 and 5 W cm$^{-2}$. As readout, mEos3.2 was excited by a 561 nm laser line operating at an average power between 1 and 2 kW cm$^{-2}$. The emission was collected by a quad band (Quad405/488/561/647 filter dual cSTORM). The exposure time was 20 ms and 4000–5000 images were typically acquired for each set of images. Images were captured using a sCMOS Flash 4.0 v3 (Hamamatsu) camera.

Note: we also imaged UTI89 cells fixed with 2% paraformaldehyde using standard protocols[55], but we noticed that the fixative seemed to interfere with protein localization. This loss of localization was not noticed when imaging live cells, similar to what previous studies with other cell division proteins have found[56,57].

**Image analysis**. Epi-fluorescence images and movies were visualized and analysed in Fiji[58]. Images and movies were post-processed for background subtraction (rolling ball radius 36 pixels). PALM images were processed using the Nikon N-STORM software and Fiji plugin ThunderSTORM[59] using Gaussian blur of 20 nm for visualization.

**Cell dimensions of rods and filaments during reversal**. Cell lengths for rods that had not been through an infection were extracted from bright-field images of respective strains. Filaments were followed for one to four divisions on agarose pads using bright field imaging. To simplify the analysis, we only followed cells pinching off from one side of a filament. 'Birth' was determined from the first image frame where the poles of cells were not connected. 'Lengths at birth' and 'symmetry at birth' statistics were generated from length measurements in MicrobeJ[60] or from manually traced lengths in Fiji. In bright field, a filament was manually classified as dead when it was clearly translucent and had similar grey values as the background. Fluorescence profiles were generated in Fiji and Full Width at Half Maxima (FWHM) was extracted by fitting a Gaussian to the generated curves. Plots and statistics were generated using the web apps PlotsOfData[61] and SuperPlotsOfData[62], as well as Origin9 pro (Origin Lab, US). Figures were prepared using Adobe Illustrator.

**Western blotting**. A volume of cells corresponding to OD$_{600}$ 0.1 was collected from cell cultures. The samples were suspended in loading buffer and resolved by sodium dodecyl sulfate-polyacrylamide gel electrophoresis. Proteins were transferred to nitrocellulose membranes using a semi-dry Transfer-Blot apparatus (Bio-

Rad). The membranes were blocked in 5% (w/v) milk and probed with anti-sera to DamX[38] (1:5000) or FtsZ (1:3333) (Agrisera).

**Statistical analysis**. All statistical analysis were performed in either Origin Pro 9 or GraphPad Prism 9. $P$ values are from unpaired two-tailed $t$ tests, and are scored as $P > 0.05$ = not statistically significant, $*P < 0.05$, $**P < 0.005$ and $***P < 0.0005$ if nothing else is indicated in the text.

**Reporting summary**. Further information on research design is available in the Nature Research Reporting Summary linked to this article.

## Data availability

The data generated in this study are provided either in the Supplementary Information or in the Source Data file. Source data are provided with this paper.

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

## Acknowledgements

The authors would like to thank David Weiss (University of Iowa) for sharing the DamX antisera and the BW25113Δ*damX* strain. Jakob Møller-Jensen is acknowledged for kindly sharing the UTI89Δ*damX* strain. The authors are grateful to Dr. Tamika Blair for initial help with the infection model. The authors acknowledge support and equipment from the UTS Microbial Imaging Facility (MIF) and a UTS infrastructure grant for a Nikon N-STORM single-molecule microscope. B.S. was supported by a CPDRF scheme from the University of Technology Sydney, and I.G.D. was supported by an Australian Research Council Future Fellowship (FT1601000010).

## Author contributions

B.S. and I.G.D. conceptualized the study. D.O.D. and M.P. contributed reagents and advice on experiments. B.S. and M.J.P. carried out all experiments. B.S. led the data analysis with input from I.G.D. B.S. and I.G.D. wrote the paper with input from D.O.D. and M.J.P.

## Competing interests

The authors declare no competing interests.
