## [Peer review file · Nature Communications]

REVIEWER COMMENTS

Reviewer #1 (Remarks to the Author):

Soderstrom et al. report that UPEC cells can grow filamentous under conditions of infection and this phenotype enhances their survival. Eventually, the filaments disperse into shorter cells. The protein DamX appears to have a role in these effects. DamX does not localize initially at future division sites in the filaments, but then localizes in ring-like structures prior to division and remains at the division sites until the daughter cells are separated. The authors analysed the filamentation phenotype and determined the localization and dynamics of DamX rings very thoroughly. However, my main point about this paper is the lack of any mechanistic insights. The authors present a wealth of empirical data but no mechanism to explain why DamX does not localize to division sites in the filaments, and how it eventually localizes and functions to trigger cell division. Hence, the findings are limited to a thoroughly analysed, interesting phenotype.

Main points:

1. The manuscript lacks mechanistic insights into how DamX might "regulate" filamentation of UPEC strains. What conditions or factors are important for this regulation and how does it work?
2. What is the evidence that DamX is the a "regulator" of cell division/dissipation? Clearly, the absence of DamX causes filamentation but not every essential cell division protein is a cell division "regulator". Could there be other proteins that prevent DamX from localizing at future division sites, hence preventing cell division? For example, do these filaments have proto-rings with FtsZ, FtsA and ZipA? Do cells filament or can filaments dissipate when FtsZ, FtsA or ZipA levels are increased? The lack of stable proto-rings could explain the filamentation phenotype, but then DamX would not be a "regulator". Is it possible that filaments (somehow) assemble stable proto-rings that then recruit DamX and all other cell division proteins? In my view they should provide evidence for the regulatory role of DamX.
3. How general are the findings? Do other UPEC strains and other E. coli strains show the same DamX-related phenotypes?
4. In my view they should analyse a damX mutant in their UTI model and include the results into the manuscript.

Minor points:

1. Line 135. It is not clear why that fact that viable filaments are shorter indicates that extensive filamentation is a stress response? A cellular stress response should increase viability under stress, not reduce it.
2. Line 242. Should it read Figure 5A (instead of Figure 4A)?
3. From line 279, OGDAlabel. What is the evidence that OGDAlabel intensity reports on sPG synthesis? To my knowledge, PG hydrolases are also active at cell division sites and potentially the label can be removed

by PG hydrolases, which cleave sPG, or by DD-carboxypeptidases that remove the (modified) D-alanine at position 5. Hence, the label intensity is the result of sPG synthesis and hydrolysis.

4. Line 433. Please add the source of OGDA.

5. Line 812. They state that cells in panel c show fluorescence staining "only at midcell". However, this must be incorrect as the whole cell body (including the sidewall) shows fluorescence in Fig. 5c.

Reviewer #2 (Remarks to the Author):

This manuscript by Söderström, Daley, and Duggin addresses the localization of the SPOR-domain-containing divisional protein DamX in filamentous uropathogenic *E. coli*. Previous studies have shown that uropathogenic *E. coli* undergoes variable filamentation during bladder infection, but the mechanisms underlying filamentation or subsequent resumption of cytokinesis are not well characterized. Separately, other general division blocks that cause filamentation in *E. coli* have been studied, as has resumption of normal division after block removal. The authors' results here show that DamX localization to sites of division are lost in filaments, but reassemble at multiple division sites during recovery (reversal). Most notably, several of these division sites within one filament appeared to constrict during reversal, and these most often occurred near the pole, in contrast to results previously obtained during recovery from other division blocks such as through antibiotic treatment.

The authors frame their study around DamX localization and behavior in filaments, and during recovery, based on two possible models where filaments contain regular DamX division rings that only become functional during reversal, or that DamX becomes dispersed in filaments and relocalizes to division sites during reversal. Based on pre-existing data that DamX requires FtsZ for localization, I would suspect that the most likely hypothesis is that FtsZ fails to form a ring in filaments and therefore DamX is dispersed - as the authors see. I find it odd that they would think DamX would be the point of regulation for controlling its functionality.

Even if theoretically possible, previous data and observations make regulation at the level of FtsZ far more likely and reasonable. It would have been more notable and of greater interest to verify that FtsZ fails to form rings in the uropathogenic context. DamX is likely secondary and of less importance. Indeed, the authors statement that their findings suggest a 'mechanism of regulated dissipation of DamX, leading to division arrest...' is unlikely. The most likely explanation is that it's FtsZ. Once FtsZ assembles at new sites, other division proteins follow suit, including DamX. There is no reason to believe things would be different in this instance of filamentation than for others where Z localization is lost. If the authors want to argue it is distinct, they really need to show that FtsZ and other division proteins

remain, but DamX is not. Only that would really argue that DamX activity is what is of primary regulatory importance here.

The previous study the authors cite in lines 315 - 319 (38) employ activation of filamentation and recovery from filamentation using experimentally-controlled switches. This paper arguably uses a more in vivo relevant 'natural' system from infection and dispersal where the cells are less uniform in growth states. This may in part account for differences they note from those previous studies regarding how many division sites are used in a recovering filament. In lines 317-318 they speak of: "division ring assemble evenly along the cell, sometimes only transiently and then assembling at new locations". It is important to note that this was localization of a fluorescently-tagged FtsA in the paper they are referring to. Not DamX or what they are looking at here; so comparison can't really be made in that regard.

The authors data highlights elongation rates compared initial filament lengths, however they don't make clear whether they also see that longer viable cells divide sooner than smaller, as previously reported in their reference (38)?

In conclusion, while there is novel data here, the majority applies to a straw-man hypothesis of DamX regulation when FtsZ regulation appears more likely (and not ruled out here.) Nonetheless, possible differences in division site usage compared to previous studies would be of interest to report, but further discussion and emphasis in relation to this would help, and I would recommend revision.

Reviewer #3 (Remarks to the Author):

The manuscript describes the development and implementation of an assay to observe filamentation recovery by uropathogenic *E. coli* (UPEC). Briefly, UPEC grows as long filaments during infection of bladder epithelial cells in humans. However, in order to infect new cells, these same filaments must revert back to short rods by a process that is not well-understood. Here, the authors show the events that occur during filamentation recovery. In particular, the authors show that UPEC forms filaments of varying lengths once inside host cells and that, upon release, shorter variants are more likely to revert to short rods. In addition, while the time to first division for a filament could be quite long, subsequent rounds of division became much shorter as cells "normalized". Similar effects were also seen for the nucleoid, which became more uniform over successive generations. Finally, the authors monitored DamX localization during filament recovery. Interestingly, the authors observed that DamX localization correlated with the onset of cell division. This observation is noteworthy because DamX is also required for infection-induced filamentation. This finding suggests that DamX plays a role in filament formation and reversion. Evidence for cell wall synthesis at sites of division was also presented.

The manuscript is well-written and the conclusions are (mostly) appropriate. The fluorescence microscopy was especially well-done and the development of a filament recovery assay paves the way to answer future mechanistic questions. For example, what are the genetic requirements for filament recovery or why do some filaments recover but not others? I have only minor comments.

Lines 2-3: "Dynamic localisation of DamX regulates bacterial filamentation and division during UPEC dispersal from host cells." The authors present data showing that DamX localizes at the start of division in filaments. However, whether DamX actively regulates filament recovery is not shown. It may be that DamX is dispensable for cell division like in wild type cells (PMID: 19684127 and 19880599).

Lines 76-77: "Whilst the exact molecular function(s) of the SPOR domains are yet to be elucidated...". Rather than focusing on what is not known about SPOR domain proteins, the authors should focus on what is known. For example, evidence suggests that FtsN triggers constriction (PMID: 19684127 and 25496259), RlpA appears to have lost its catalytic activity in *E. coli* (PMID: 24806796 and 31286580), and DamX and DedD stimulate the enzymatic activity of cell wall synthases (PMID: 33144379). The fact that DamX stimulates cell wall synthesis supports a role for DamX in filament recovery (e.g., Fig. 5).

Lines 133-134: "Therefore, most of the viable filaments were significantly shorter than the average..." The authors state that longer filaments were less viable than shorter ones. Thus, it seems a little strange to show the opposite in Supplementary Fig. 3C.

Lines 331-333: "Surprisingly, we instead observed that fluorescently-tagged DamX (mEos3.2-DamX) was not initially assembled at potential division sites in UPEC filaments..." It would be helpful to the reader if the authors explained the logical conclusion for this statement. Namely, that DamX may be required for filament formation and recovery.

Discussion: Since the authors have essentially developed a shape recovery assay, it would be helpful to compare their findings to other systems. In particular, the spheroplast recovery assay. Without going into detail, there are a lot of parallels between filament and spheroplast recovery (PMID: 16589911 and 23543719). Importantly, spheroplasts are thought to be the reservoir for recurrent UTI (PMID: 31558767).

Supplementary Figure 1C. Suggest adding molecular weight markers to the blot and noting the sizes for DamX and mEos3.2-DamX in the figure legend.

REVIEWER COMMENTS

Reviewer #1 (Remarks to the Author):

Soderstrom et al. report that UPEC cells can grow filamentous under conditions of infection and this phenotype enhances their survival. Eventually, the filaments disperse into shorter cells. The protein DamX appears to have a role in these effects. DamX does not localize initially at future division sites in the filaments, but then localizes in ring-like structures prior to division and remains at the division sites until the daughter cells are separated. The authors analysed the filamentation phenotype and determined the localization and dynamics of DamX rings very thoroughly. However, my main point about this paper is the lack of any mechanistic insights. The authors present a wealth of empirical data but no mechanism to explain why DamX does not localize to division sites in the filaments, and how it eventually localizes and functions to trigger cell division. Hence, the findings are limited to a thoroughly analysed, interesting phenotype.

AU: We thank the reviewer for taking the time to review our manuscript. We have addressed the comments in a point-by-point manner below.

Main points:

1. The manuscript lacks mechanistic insights into how DamX might "regulate" filamentation of UPEC strains. What conditions or factors are important for this regulation and how does it work?

AU: This and related comments from the other reviewers have been the main focus of our further experimentation and revisions. To identify the main mechanism of cell division arrest leading to infection-related filamentation, we have added new results on the dynamic behaviour of FtsZ during filamentation and the importance of DamX in regulating bacterial physiology during the filamentation stage in our infection model. Interestingly, this revealed that the mechanism of cell division regulation occurs at the stage of FtsZ ring stabilization – i.e. not initial ring assembly nor later stages that have been recognised by the cell division field. We believe this represents one of the first key insights into how infection related filamentation occurs. The FtsZ result has a more general implication for cell division too; that the formation of an FtsZ-ring is not only the deciding factor in the cellular 'decision' to divide or not, because we discovered that the rings can be unstable and continually reappear in other locations in the filament.

We also confirmed that a UTI89 Δ damX strain is unable to filament in our cellular UTI model, and we discovered that it affects bacterial cell integrity. These are the most substantial phenotypes identified for DamX to date. These confirm that DamX is a critical part in the regulatory mechanism controlling filamentation and survival during the dispersal stage of cellular infection, and that it also plays a significant role in the filament and vegetative cell division processes. By complementing the UTI89 Δ damX strain with mEos3.2-DamX, we were able to rescue the filamentation phenotype (this also showed that the protein fusion is functional).

We have also added new text in the discussion clarifying why the results indicate that DamX is a key part of the mechanism responsible for regulating filamentation. Lines 443 – 445.

New FtsZ data in Figure 3.

New DamX data in Figure 4, with supporting new data in Supplementary Figure 5.

See further details below.

2. What is the evidence that DamX is the a "regulator" of cell division/dissipation? Clearly, the absence of DamX causes filamentation but not every essential cell division protein is a cell division "regulator". Could there be other proteins that prevent DamX from localizing at future division sites, hence preventing cell division? For example, do these filaments have proto-rings with FtsZ, FtsA and ZipA? Do cells filament or can filaments dissipate when FtsZ, FtsA or ZipA levels are increased? The lack of stable proto-rings could explain the filamentation phenotype, but then DamX would not be a "regulator". Is it possible that filaments (somehow) assemble stable proto-rings that then recruit DamX and all other cell division proteins? In my view they should provide evidence for the regulatory role of DamX.

AU: We agree there are very likely to be several other players involved in the filamentous response pathway that DamX is a key part of, and our laboratories are searching for these. However, it is important to note that DamX is strictly required for filamentation and our results confirm that – that is, its deletion completely prevents infection-related filamentation but not bacterial proliferation. Therefore DamX is a key part of the regulation of division during infection-related filamentation. But how it regulates this process is unknown and we begin to address this here. We believe our study will help this filamentous response to become a model of bacterial physiological responses during infection. We have now explained this function of DamX more clearly in the revised version of the paper and have added additional findings that support its important role during infection,

We also thank the reviewer for the suggestion of initially looking at a proto-ring member. As introduced above, we looked at the behaviour of FtsZ(-mCitrine) in filaments after the infection. We found that FtsZ localization and dynamics are not 'final', in the sense that a Z-ring often formed but then disassembled quickly after, without resulting in a division event.

These results are described in detail in a new results section on lines 222 - 246, in the discussion at lines 419-435, and in a new Figure 3.

3. How general are the findings? Do other UPEC strains and other *E. coli* strains show the same DamX-related phenotypes?

AU: As noted above, the FtsZ results now included can be generalized to the level of bacterial cell division and the principles of a cell's commitment to divide. Similarly, the DamX phenotypes are also related to general functions in bacterial cell division and filamentation – this is an apparent case of a division related protein with two opposing roles depending on conditions, and we speculate that this 'switching/opposing roles' concept may apply to other proteins more bacterial response pathways too. In the revised manuscript, we have compared the $\Delta damX$ phenotypes in a standard non-pathogenic *E. coli* model strain (K12 - BW25113) with the UPEC UTI89 strain. Interestingly, we found an elongation phenotype for the UTI89 $\Delta damX$ strain that was not detected in non-pathogenic *E. coli* K-12 in a previous study (Ardes et al J. Bact. 2010), in agreement with our results on K-12 here. We also show that this phenotype can be reversed by complementing the UTI89 $\Delta damX$ strain with DamX produced from a plasmid. The implications of these wild-type strain differences are potentially profound with regards to the claims made in other studies to the generality of studying the *E. coli* K-12 strain. Time will tell if it is atypical, through further studies on different strains beyond the focus and scope of the current work, but we expect our study will instigate further investigations into other important differences between the function of genes in the context of other wild-type strains or pathotypes, even in processes as fundamental as cell division.

These new results are described on lines 248- 276 and added in a new Figure 4 with supplementary data in a new Supplementary Figure 5.

4. In my view they should analyse a damX mutant in their UTI model and include the results into the manuscript.

AU: Upon the suggestion of the reviewer, we have now included an analysis of a UTI89 $\Delta damX$ strain in the UTI model.

As described previously (Khandige, S., DamX Controls Reversible Cell Morphology Switching in Uropathogenic Escherichia coli. mBio. 2016), we too found that the UTI89 $\Delta damX$ strain was not able to produce filaments. We could however rescue the filamentation phenotype by complement the UTI89 $\Delta damX$ strain with mEos3.2-DamX protein fusion produced from a plasmid. And we made additional important observations into the function of DamX that confirm and substantially expand on the previous finding. These results are discussed on lines 277 - 290 and included in the new Figure 4.

Minor points:

1. Line 135. It is not clear why that fact that viable filaments are shorter indicates that extensive filamentation is a stress response? A cellular stress response should increase viability under stress, not reduce it.

AU: We agree with the reviewer and deleted that last part of the sentence. Line 146.

2. Line 242. Should it read Figure 5A (instead of Figure 4A)?

AU: Figure 6A shows multiple generations of mEos3.2-DamX rings. To avoid any confusion, we have indicated specifically that we mean mEos3.2-DamX rings on line 333.

3. From line 279, OGDA label. What is the evidence that OGDA intensity reports on sPG synthesis? To my knowledge, PG hydrolases are also active at cell division sites and potentially the label can be removed by PG hydrolases, which cleave sPG, or by DD-carboxypeptidases that remove the (modified) D-alanine at position 5. Hence, the label intensity is the result of sPG synthesis and hydrolysis.

AU: We have added that OGDA intensity can be the result of either/both sPG synthesis and hydrolysis. Lines 365, 367, 372.

4. Line 433. Please add the source of OGDA.

AU: Line 566. We have moved the reference for OGDA to the correct place.

5. Line 812. They state that cells in panel c show fluorescence staining "only at midcell". However, this must be incorrect as the whole cell body (including the sidewall) shows fluorescence in Fig. 5c.

AU: Line 1022. We acknowledge that we were not 100% accurate in our description. We have changed the text accordingly (Now Fig. 7c).

Reviewer #2 (Remarks to the Author):

This manuscript by Söderström, Daley, and Duggin addresses the localization of the SPOR-domain-containing divisional protein DamX in filamentous uropathogenic *E. coli*. Previous studies have shown that uropathogenic *E. coli* undergoes variable filamentation during bladder infection, but the mechanisms underlying filamentation or subsequent resumption of cytokinesis are not well characterized. Separately, other general division blocks that cause filamentation in *E. coli* have been studied, as has resumption of normal division after block removal. The authors' results here show that DamX localization to sites of division are lost in filaments, but reassemble at multiple division sites during recovery (reversal). Most notably, several of these division sites within one filament appeared to constrict during reversal, and these most often occurred near the pole, in contrast to results previously obtained during recovery from other division blocks such as through antibiotic treatment.

AU: We thank the reviewer for taking the time to review our manuscript. We have addressed the comments in a point-by-point manner below.

The authors frame their study around DamX localization and behavior in filaments, and during recovery, based on two possible models where filaments contain regular DamX division rings that only become functional during reversal, or that DamX becomes dispersed in filaments and relocalizes to division sites during reversal. Based on pre-existing data that DamX requires FtsZ for localization, I would suspect that the most likely hypothesis is that FtsZ fails to form a ring in filaments and therefore DamX is dispersed - as the authors see. I find it odd that they would think DamX would be the point of regulation for controlling its functionality.

AU: As noted in our response to reviewer 1 (point 2), as previously observed (Khandige, S. , DamX Controls Reversible Cell Morphology Switching in Uropathogenic Escherichia coli. mBio. 2016), the DamX knockout strain was not able to produce filaments, but the cells were able to proliferate and divide quite well. This is a strong indication that DamX is indeed involved in the regulation of filamentation. Our new results confirm and extend this. Thus our conclusion regarding DamX function is drawn directly from the findings rather than a pre-conceived hypothesis. We too would not have predicted this beforehand.

These data are included in a new Figure 4.

Even if theoretically possible, previous data and observations make regulation at the level of FtsZ far more likely and reasonable. It would have been more notable and of greater interest to verify that FtsZ fails to form rings in the uropathogenic context. DamX is likely secondary and of less importance. Indeed, the authors statement that their findings suggest a 'mechanism of regulated dissipation of DamX, leading to division arrest...' is unlikely. The most likely explanation is that it's FtsZ. Once FtsZ assembles at new sites, other division proteins follow suit, including DamX. There is no reason to believe things would be different in this instance of filamentation than for others where Z localization is lost. If the authors want to argue it is distinct, they really need to show that FtsZ and other division proteins remain, but DamX is not. Only that would really argue that DamX activity is what is of primary regulatory importance here.

AU: We agree and have done this, as described in detail above (reviewer 1, point 1). These results are described on lines 222 - 248, and in a new Figure 3.

New text in the discussion is added on lines 419 – 435, 443 – 445 and 454-464 to address these findings.

The previous study the authors cite in lines 315 - 319 (38) employ activation of filamentation and recovery from filamentation using experimentally-controlled switches. This paper arguably uses a more in vivo relevant 'natural' system from infection and dispersal where the cells are less uniform in growth states. This may in part account for differences they note from those previous studies regarding how many division sites are used in a recovering filament. In lines 317-318 they speak of: division ring assemble evenly along the cell, sometimes only transiently and then assembling at new locations". It is important to note that this was localization of a fluorescently-tagged FtsA in the paper

they are referring to. Not DamX or what they are looking at here; so comparison can't really be made in that regard.

AU: We agree and have clarified this comparison in the new text Lines 412 - 435. We also note that our new results with FtsZ make these systems more comparable, albeit with different proto-ring proteins (FtsZ/FtsA).

The authors data highlights elongation rates compared initial filament lengths, however they don't make clear whether they also see that longer viable cells divide sooner than smaller, as previously reported in their reference (38)?

AU: We looked at the length of the filaments at start of imaging vs the time to first division and found not apparent correlation. Figure 2f. Line 173-175.

In conclusion, while there is novel data here, the majority applies to a straw-man hypothesis of DamX regulation when FtsZ regulation appears more likely (and not ruled out here.) Nonetheless, possible differences in division site usage compared to previous studies would be of interest to report, but further discussion and emphasis in relation to this would help, and I would recommend revision.

AU: We agree that we had not ruled out regulation of FtsZ, and our original paper did not intend to suggest this. We have now been able to comment on this in the paper because we have focussed substantial new experimentation on it, as outlined above.

It is important to clarify that we are not proposing that DamX's moderate function in division being regulated (inhibited) to arrest division and cause filamentation. Rather, our new data and some previous published data show that DamX is actually effecting the regulation of division through an additional function. This might indeed be a primary function of DamX, which is still not well understood. The key data comes from studies of the DamX knockout strain. This demonstrated that DamX is strictly required for infection-related filamentation, but is dispensible for bacterial growth and division in the infections. Its target in the division machinery is not precisely known, but our new results would suggest it affects Z-ring stability, as predicted by the reviewer. We have endeavoured to clarify the manuscript throughout on this point, to avoid similar reader misunderstanding.

As noted above our new data do support the view that division is controlled at the specific stage of Z-ring stabilization, leading to infection-related filamentation. The simplest corollary of our results is that DamX somehow directly causes the instability of FtsZ rings during filamentation from its position all throughout the cell membrane. Then, when filaments switch from further cell elongation to initiate division, DamX switches function for its role in division, and our localization data are consistent with this. The exact molecular players and how they are regulated at the molecular level are the subject of a lot of future work in what we anticipate will become a model of bacterial infection-related response.

Reviewer #3 (Remarks to the Author):

The manuscript describes the development and implementation of an assay to observe filamentation recovery by uropathogenic *E. coli* (UPEC). Briefly, UPEC grows as long filaments during infection of bladder epithelial cells in humans. However, in order to infect new cells, these same filaments must revert back to short rods by a process that is not well-understood. Here, the authors show the events that occur during filamentation recovery. In particular, the authors show that UPEC forms filaments of varying lengths once inside host cells and that, upon release, shorter variants are more likely to revert to short rods. In addition, while the time to first division for a filament could be quite long, subsequent rounds of division became much shorter as cells “normalized”. Similar effects were also seen for the nucleoid, which became more uniform over successive generations. Finally, the authors monitored DamX localization during filament recovery. Interestingly, the authors observed that DamX localization correlated with the onset of cell division. This observation is noteworthy because DamX is also required for infection-induced filamentation. This finding suggests that DamX plays a role in filament formation and reversion. Evidence for cell wall synthesis at sites of division was also presented.

AU: We thank the reviewer to taking the time to review our manuscript. We have addressed the comments in a point-by-point manner below.

The manuscript is well-written and the conclusions are (mostly) appropriate. The fluorescence microscopy was especially well-done and the development of a filament recovery assay paves the way to answer future mechanistic questions. For example, what are the genetic requirements for filament recovery or why do some filaments recover but not others? I have only minor comments.

AU: We appreciate the support and note that we are working on identifying the other mechanistic and genetic requirements for filamentation and reversal.

We do not yet understand why some cells die and others survive, but the conditions are clearly a type of stress response

These new data are included as a new results subsection, lines 222-246 and a in new Figure 3.

Lines 2-3: “Dynamic localisation of DamX regulates bacterial filamentation and division during UPEC dispersal from host cells.” The authors present data showing that DamX localizes at the start of division in filaments. However, whether DamX actively regulates filament recovery is not shown. It may be that DamX is dispensable for cell division like in wild type cells (PMID: 19684127 and 19880599).

AU: We have added new (surprising) data on UTI89 Δ damX showing a elongated phenotype for > 10% of the cell population. This is in contrast to previous studies on Δ damX deletions in non-pathogenic strains (PMID: 19684127 and 19880599), where no clear phenotype was found in single knockouts (we confirmed these data in the current manuscript). Especially Supplementary Movie 9.

We ran the infection model using both UTI89 Δ damX and UTI89 Δ damX complemented with a mEos3.2-DamX from a plasmid. UTI89 Δ damX did not produce filaments (as expected, and seen previously: Khandige, S. , DamX Controls Reversible Cell Morphology Switching in Uropathogenic Escherichia coli. mBio. 2016). We could rescue the filamentation phenotype in the complemented strain, indicating the importance of DamX for filamentation of UPEC during UTIs.

These new data are included in a new section in the manuscript. Lines 248 – 290.

We have also added a new Figure 4 on this, with supplementary data in a new Supplementary Figure 5.

Lines 76-77: “Whilst the exact molecular function(s) of the SPOR domains are yet to be elucidated...”. Rather than focusing on what is not known about SPOR domain proteins, the authors should focus on what is known. For example, evidence suggests that FtsN triggers constriction (PMID: 19684127 and

25496259), RlpA appears to have lost its catalytic activity in *E. coli* (PMID: 24806796 and 31286580), and DamX and DedD stimulate the enzymatic activity of cell wall synthases (PMID: 33144379). The fact that DamX stimulates cell wall synthesis supports a role for DamX in filament recovery (e.g., Fig. 5).

AU: We thank the reviewer for their thoughtful insights into the functions of the SPOR domain proteins and have change the text accordingly. Lines 78 - 87.

We have clarified the text at lines 450 - 453, in regards to the possible role for DamX during cell wall synthesis.

Lines 133-134: "Therefore, most of the viable filaments were significantly shorter than the average..." The authors state that longer filaments were less viable than shorter ones. Thus, it seems a little strange to show the opposite in Supplementary Fig. 3C.

AU: We thank the review for this comment. Supplementary figure 3C was intended to help reader visually see green as live and magenta as dead cells, but we realise that we already made the point in Figure 1d and e. Therefore, in order not to confuse readers we have removed the images for Supplementary figure 3C altogether.

Lines 331-333: "Surprisingly, we instead observed that fluorescently-tagged DamX (mEos3.2-DamX) was not initially assembled at potential division sites in UPEC filaments..." It would be helpful to the reader if the authors explained the logical conclusion for this statement. Namely, that DamX may be required filament formation and recovery.

AU: We have added a sentence clarifying the reasoning behind the statement. Line 443 - 445.

Discussion: Since the authors have essentially developed a shape recovery assay, it would be helpful to compare their findings to other systems. In particular, the spheroplast recovery assay. Without going into detail, there are a lot of parallels between filament and spheroplast recovery (PMID: 16589911 and 23543719). Importantly, spheroplasts are thought to be the reservoir for recurrent UTI (PMID: 31558767).

AU: We understand the view of the reviewer, but we believe that this is may be outside the scope of this paper and would only confuse a potential reader. We have added text regarding strain differences in general on lines 412-418.

Supplementary Figure 1C. Suggest adding molecular weight markers to the blot and noting the sizes for DamX and mEos3.2-DamX in the figure legend.

AU: Done.

REVIEWERS' COMMENTS

Reviewer #1 (Remarks to the Author):

The authors have addressed all of my points and revised the manuscript accordingly. In my view the manuscript has much improved by the revision, in particular through the addition of new data on FtsZ and the damX mutant. I have no further points on the revised manuscript and want to congratulate the authors to this nice piece of work.

Reviewer #3 (Remarks to the Author):

Lines 436-464: At this point, it would be helpful for the authors to make a broader statement about the role of DamX in shape shifting. A closer look at the literature reveals that damX (VPA1294) expression is upregulated >100-fold during swarmer cell differentiation in *Vibrio parahaemolyticus* (PMID: 21166906). Like IRF, swarmer cell differentiation involves filamenting. It may be that DamX is generally employed by bacteria as a defense strategy.

REVIEWERS' COMMENTS

Reviewer #1 (Remarks to the Author):

The authors have addressed all of my points and revised the manuscript accordingly. In my view the manuscript has much improved by the revision, in particular through the addition of new data on FtsZ and the damX mutant. I have no further points on the revised manuscript and want to congratulate the authors to this nice piece of work.

AU: We thank the reviewer for their assessment of our revised manuscript and we are pleased to hear that they are now overall satisfied.

Reviewer #3 (Remarks to the Author):

Lines 436-464: At this point, it would be helpful for the authors to make a broader statement about the role of DamX in shape shifting. A closer look at the literature reveals that damX (VPA1294) expression is upregulated >100-fold during swarmer cell differentiation in *Vibrio parahaemolyticus* (PMID: 21166906). Like IRF, swarmer cell differentiation involves filamenting. It may be that DamX is generally employed by bacteria as a defense strategy.

AU: We thank the reviewer for examining our manuscript once more.

On the reviewers request, we have added a stament on the possible role of DamX in morphology regulation.

We have also added the reference (PMID: 21166906).

Lines 460-462 in the discussion.